# Changes in the mRNA expression of glycolysis-related enzymes of *Candida albicans* during inhibition of intramitochondrial catabolism under anaerobic condition

Ken Okabayashi[1], Hitomi Ogawa[1], Yuto Hirai[1], Kureha Nagata[1], Yukiko Sato[1], Takanori Narita[1]*, Kazuo Satoh[2,3], Koichi Makimura[2,3]

**1** Department of Veterinary Medicine, College of Bioresource Sciences, Nihon University, Fujisawa, Kanagawa, Japan, **2** Medical Mycology, Space and Environmental Medicine, Graduate School of Medicine/Medical Technology, Teikyo University, Tokyo, Japan, **3** General Medical Education and Research Center, Teikyo University, Tokyo, Japan

* narita.takanori@nihon-u.ac.jp

**Data Availability Statement:** All relevant data are within the paper.

## Abstract

*Candida albicans* can cause two major types of infections: superficial infection and systemic candidiasis. *C. albicans* infects diverse host niches, owing to a wide range of virulence factors and attributes, such as morphological transitions and phenotypic switching. *C. albicans* uses glycolysis, followed by alcoholic fermentation or mitochondrial respiration to rapidly generate ATP under aerobic conditions. In this study, we quantified the mRNA expression of several glycolysis-related enzymes associated with the initial phase of environmental changes using two strains: a type strain, NBRC 1385, and a strain from a patient with auto-brewery syndrome, LSEM 550. Additionally, we analyzed the regulation of a rate-limiting enzyme in glycolysis, phosphofructokinase 1 (PFK1). Our results showed that the mRNA expression of enzymes in the middle and last stages of glycolysis and alcoholic fermentation increased, and that of mitochondrial respiration enzymes decreased under short-term anaerobic conditions. Carbonyl cyanide-p-trifluoromethoxyphenylhydrazone (FCCP) administration showed similar results under anaerobic conditions. Moreover, PFK1 maintained its regulatory effect under different conditions; no significant change was observed in its mRNA expression. Our results suggest that *C. albicans* obtains energy via carbohydrate catabolism in the early phase of environmental change and survives in various parts of the host.

## Introduction

The opportunistic human fungal pathogen *Candida albicans* is a causative yeast in both superficial and systemic infections. Systemic infections are associated with a high mortality rate, whereas superficial infections are non-lethal [1]. *C. albicans* is a polymorphic fungus, which can grow either as an ovoid-shaped budding yeast, as elongated ellipsoid cells with

**Funding:** The authors received no specific funding for this work.

**Competing interests:** The authors have declared that no competing interests exist.

constrictions at the septa (pseudohyphae), or as parallel-walled true hyphae [2]. *C. albicans* can infect varied host niches, because of various virulence factors and fitness attributes [1,3]. In addition, mild heat stress causes morphological changes in the cell wall structure and biofilm formation by regulating glycolytic gene expression in the early phase of heat stress [4,5].

*C. albicans*, a facultative aerobe, metabolizes carbon sources under aerobic conditions [6]. Glycolysis is responsible for converting glucose into the key metabolite pyruvate while also producing ATP and NADH. As a central metabolic pathway, glycolysis is strictly regulated and provides ATP more rapidly than the tricarboxylic acid (TCA) cycle in the mitochondria; however, it produces lower levels of ATP [7]. The enzyme glucokinase 1 (GLK1) catalyzes the first step of glycolysis to phosphorylate glucose at C6. Phosphofructokinase 1 (PFK1), the most important rate-limiting enzyme in glycolysis, catalyzes fructose 6-phosphate to fructose 1,6-bisphosphate. Glyceraldehyde 3-phosphate dehydrogenase (TDH3; GAPDH) regulates the sixth step (intermediate step) of glycolysis by catalyzing glyceraldehyde 3-phosphate to 1,3-bisphosphoglycerate. Pyruvate kinase (CDC19) catalyzes the conversion of phosphoenol-pyruvate to pyruvate, which is the final step in glycolysis. Additionally, alcohol dehydrogenase 1 (ADH1) catalyzes the reduction of acetaldehyde to ethanol during alcohol fermentation, and pyruvate carboxylase 2 (PYC2) catalyzes pyruvate to oxaloacetate. Pyruvate dehydrogenase, which is involved in acetyl-CoA biosynthesis from pyruvate, contains several components: E1 α subunit (PDA1), E1 β subunit (PDB1), and E3-binding protein (PDX1). *C. albicans* employs two major strategies for energy production: anaerobic fermentation, which produces ethanol, and mitochondrial respiration, which produces additional ATP via the TCA cycle and oxidative phosphorylation. However, intramitochondrial catabolism would be inhibited by anaerobic environment or protonophore administration such as carbonyl cyanide-*p*-trifluoromethoxyphenylhydrazone (FCCP) [8]. The regulation of glycolysis in *C. albicans* during cell growth and morphological switching under various environmental conditions that favor high rates of glycolysis is unclear.

Furthermore, alcohol, a metabolite of fermentation by yeast, induces toxicity in various organisms, including humans. Chronic alcohol consumption may cause alcoholism and is associated with various diseases, including psychiatric symptoms, malnutrition, chronic pancreatitis, alcoholic liver disease, hepatocellular carcinoma, and coronary heart disease [9]. In auto-brewery syndrome with pathogenic yeast strain with high alcohol fermentation ability, alcohol is generated within the host body via anaerobic fermentation [10]. In this case, there are two types of symptoms, i.e., those associated with long-term alcohol exposure and the ones typically associated with pathogenic yeast.

This study analyzes the effects of morphological and environmental changes on energy metabolism in *C. albicans* by quantifying the mRNA expression of enzymes associated with glycolysis, intramitochondrial catabolism, and alcoholic fermentation in *C. albicans* during the stable growth period. Two strains were used in this study, of which one is a type strain, while the other is with high alcohol fermentation ability, to examine strain-associated differences in metabolism. Furthermore, we analyzed PFK1 regulation under major environmental factors.

## Methods

### Strains and media

The strains of *C. albicans* used in this study were the type strain, NBRC 1385, and the strain with high alcohol fermentation ability, LSEM 550. Both strains have been registered in the mycological culture collection of Teikyo University and preserved for a long period of time. LSEM 550 was identified as *C. albicans* by the DNA sequencing of the 26S rDNA D1/D2 region (DDBJ Accession number: LC727662), which showed a high alcohol fermentation ability.

At the start of this study, both strains (NBRC 1385 and LSEM 550) were obtained from the mycological culture collection of Teikyo University and had been maintained on diluted Sabouraud's glucose agar: 0.2% glucose, 0.1% peptone, 0.1% $KH_2PO_4$, $MgSO_4 \cdot 7H_2O$, and 2% agar at 25°C.

## Growth curve

Each strain was precultured in Sabouraud dextrose broth (SDB) containing 4% glucose and 1% peptone with shaking at 37°C overnight and subcultured into SDB (1,000 cells/μL). The number of yeast cells was counted with a hemocytometer every 3 h to generate a growth curve.

## Anaerobic culture and inhibition of mitochondrial oxidative phosphorylation

The yeast cells in the linear growth phase (NBRC 1385: 12–15 h, LSEM 550: 15–18 h) were subjected to anaerobic cultivation by Anaero Pack-Anaero (Mitsubishi Gas Chemical, Tokyo, Japan) or uncouple of mitochondrial oxidative phosphorylation by administration of 10 μM FCCP, with shaking at 37°C for 3 h.

## Quantitative reverse transcriptase polymerase chain reaction (qRT-PCR)

Total RNA was extracted from yeast samples in SDB under atmospheric or anaerobic culture conditions. To this end, yeast cells were harvested by centrifugation at $820 \times g$ and were disrupted using a sterilized and deep-frozen pestle and mortar. Total RNA was extracted from disrupted yeast cells using TRIzol reagent (Life Technologies Co., Carlsbad, CA, USA), according to the manufacturer's instructions. First-strand cDNA synthesis was performed using 500 ng of total RNA with a reverse transcription reagent kit (PrimeScript RT Master Mix; Takara Bio Inc., Kusatsu, Japan). Quantitative PCR (qPCR) assays were performed using 2 μL of first-strand cDNA in a total reaction volume of 25 μL (SYBR Premix Ex Taq™ II, Takara Bio Inc.). Primers used in this study (Table 1) were purchased from Sigma-Aldrich (St. Louis, MO, USA). The Polymerase chain reaction assay was conducted using a thermal cycler (Thermal Cycler Dice® Real-Time System II TP900, Takara Bio Inc.), using the following cycling conditions: denaturation at 95°C for 5 s, annealing, and extension at 60°C for 30 s. Upon analysis of the amplicons, we confirmed that they showed a single peak and single band in the dissociation curve and agarose gel electrophoresis. The qPCR results were analyzed using the second derivative method and comparative cycle threshold method using Thermal Cycler Dice® Real-Time System Software Ver.5.11B (Takara Bio Inc.). Housekeeping genes were evaluated based on the standard deviation of the cycle threshold for actin (ACT1), GAPDH, TATA-binding protein (TBP1), and 18S ribosomal RNA (RDN18) using the BestKeeper software [11] to select the internal standard for relative mRNA expression. The data were presented as mean ± standard error (SE) and analyzed statistically between aerobic culture, as control, and anaerobic culture or FCCP-administered culture, using Student's $t$-test.

## Partial purification of PFK1

Yeast cells were harvested from 1 L of SDB by shaking at 37°C during the linear proliferation period. All the purification steps were performed at 4°C. The yeast cells were centrifuged at $1,500 \times g$ to eliminate the supernatant and rinsed with 30 mM phosphate buffer (pH 7.5). The yeast cells suspended in 100–200 mL extraction buffer—50 mM phosphate buffer (pH 7.1), 0.5 mM EDTA, 0.5 mM phenylmethylsulfonyl fluoride, and 5 mM

**Table 1. Primers used in this study.**

| gene | Accession number | Primer sequence (5'-3') | | Length (bp) |
|------|------------------|-------------------------|---|-------------|
| TBP1 | XM_705629.1 | Forward: GGCTTAGCTTTTGCTCATGGT | | 141 |
| | | Reverse: TCTCTTTTTGGCACCCGTCA | | |
| ACT1 | XM_019475182.1 | Forward: ACGGTGAAGAAGTTGCTGCT | | 93 |
| | | Reverse: GGAAAACAGCTCTTGGAGCG | | |
| RDN18 | XR_002086442.1 | Forward: TCTTGTGAAACTCCGTCGTG | | 105 |
| | | Reverse: AGGGACGTAATCAACGCAAG | | |
| GLK1 | XM_705084.2 | Forward: ACCAACAAGGAGGAACGTGA | | 87 |
| | | Reverse: TGCAATTGGGATCGCAGACA | | |
| PFK1 | XM_716783.1 | Forward: CCAAGTGAAGTTGGCGGTGG | | 113 |
| | | Reverse: CCGGCTCTAACAACAGCACG | | |
| GAPDH | XM_714816.1 | Forward: TGCTGCTAAAGCCGTTGGTA | | 85 |
| | | Reverse: AACATCGGTGGTTGGGACTC | | |
| CDC19 | XM_709841.1 | Forward: AAAAGGCCATTGCCTACCCA | | 90 |
| | | Reverse: CGGCAACAGCACAAGTTTCA | | |
| ADH1 | XM_716812.2 | Forward: AGTTGGTGGTCACGAAGGTG | | 93 |
| | | Reverse: GATACCGGCAAAGTCACCGA | | |
| PYC2 | XM_715820.2 | Forward: TGATTCTGCCGGTACTGGTG | | 86 |
| | | Reverse: ACATGGAATTCAGAGCGGCA | | |
| PDA1 | XM_710313.1 | Forward: GGCTTCTTGACCGACAGACA | | 81 |
| | | Reverse: AGAATGGAAATGGCAGCCGA | | |
| PDB1 | XM_712005.2 | Forward: ACCAACACCAAAACCTGGGA | | 193 |
| | | Reverse: TGTTGGTCATAGTCGTGCCC | | |
| PDX1 | XM_717131.1 | Forward: GGGTTCTTCTCCTGTGGAGC | | 192 |
| | | Reverse: CATTGATGTCGAAGCAGCCG | | |

2-mercaptoethanol, with 10 mM dithiothreitol—were homogenized with quartz sand and centrifuged at $1,500 \times g$ for 5 min.

The cytosolic extract was precipitated with polyethylene glycol, based on the precipitation data of *Saccharomyces cerevisiae* PFK1 [12]. According to this method, after precipitation in a solution of 4% polyethylene glycol (PEG), the supernatant was reprecipitated in a solution of 14% PEG. The subsequent precipitate was used for the enzyme assay as partially purified PFK1, after being dissolved in 1 mL extraction buffer. The protein concentration of partially purified PFK1 was determined using the Bradford method with bovine serum albumin as the standard [13].

## PFK1 activity assay

PFK1 activity assay was conducted as previously reported [14]. The enzymatic reaction was initiated by adding partially purified PFK1 at 25°C, and the rate of NADH oxidation was monitored at 340 nm using a spectrophotometer. The activity was determined in a reaction mixture containing the following components in a final volume of 1 mL: 50 mM 4-(2-hydroxyethyl)-1-piperazineethanesulfonic acid (HEPES) buffer (pH 7.0–9.2) or 50 mM 2-(N-morpholino) ethanesulfonic acid (MES) buffer (pH 6.0–6.8), 5 mM $MgCl_2$, 0.5 mM fructose 6-phosphate (F-6-P), 3 mM ATP, 0.2 mM NADH, 1 mM $NH_4Cl$, 5 mM KH2PO4, aldolase (0.3 units), triosephosphate isomerase (0.5 units), and glycerol-3-phosphate dehydrogenase (5 units). One unit of PFK1 activity is defined as the amount of enzyme that phosphorylates 1μmol of F-6-P/min 25°C [12].

## Results

### Growth curve

To determine the linear growth phase of each strain (NBRC 1385 and LSEM 550) in SDB with shaking at 37°C, the number of yeast cells was counted every 3 h to generate a growth curve. The growth curve of NBRC 1385 was linear until 18 h (approximately 170,000 /μL) and then plateaued or slightly decreased (Fig 1A). The growth curve of LSEM 550 was roughly linear until 21 h (approximately 85,000 /μL) and then slightly decreased (Fig 1B). Therefore, we estimated that it would be appropriate to start the anaerobic culture experiments (for 3 h during the growth phase) after 12 h (approximately 100,000 /μL) and 15 h (approximately 60,000 /μL) from the start of subculture for NBRC 1385 and LSEM 550, respectively.

### Anaerobic culture and inhibition of mitochondrial oxidative phosphorylation

To evaluate the morphological changes in *C. albicans* in response to harsh environmental factors, yeast cells were cultured in anaerobic conditions or were administered FCCP to uncouple mitochondrial oxidative phosphorylation and were then observed under an optical microscope. The morphology of both strains under anaerobic or FCCP treatment conditions did not differ from that of the strains under aerobic conditions.

### Quantitative reverse transcriptase PCR

The mRNA expression of GLK1, PFK1, GAPDH, CDC19, ADH1, PYC2, PDA1, PDB1, and PDX1 under anaerobic or FCCP treatment was analyzed in both strains to determine the changes in energy production in *C. albicans* under harsh environmental conditions. Upon analysis of all amplicons by qRT-PCR, we confirmed that they showed a single band in the agarose gel electrophoresis results and a single peak in the dissociation curve. *TBP1* was selected as a stable reference gene because it had a standard deviation <1 and a larger correlation coefficient than *ACT1*, *GAPDH*, or *RDN18* (determined using the BestKeeper software) and was used to calculate relative mRNA expression as the internal standard. The results of relative mRNA expression of NBRC 1385 and LSEM 550 based on TBP1 are shown in Figs 2 and 3, respectively.

The mRNA expression of NBRC 1385 in the anaerobic culture showed higher levels of GAPDH, CDC19, and ADH1 and lower levels of PDB1 than that of the aerobic culture (control). The mRNA expression of NBRC 1385 with FCCP administration showed higher levels of GLK1 and ADH1 and lower PDB1 levels than that of the control. The mRNA expression of LSEM 550 in anaerobic culture showed higher levels of GAPDH, CDC19, and ADH1 and lower levels of GLK1, PYC2, PDA1, PDB1, and PDX1 compared with that of the control. The mRNA expression of LSEM 550 after uncouple of mitochondrial oxidative phosphorylation showed higher levels of GAPDH and ADH1 and lower levels of PYC2, PDA1, PDB1, and PDX1 compared with that of the control.

The mRNA expression in NBRC 1385 and LSEM 550 showed similar trends compared with the anaerobic culture or FCCP administration, while LSEM550 showed more significant changes than NBRC1385 (Fig 2 and 3).

### Regulation of PFK1 activity

PFK1 is the most important rate-limiting enzyme in glycolysis. Therefore, we analyzed the changes in PFK1 activity under harsh environmental factors in both strains. The specific activity of PFK1 extracted from NBRC 1385 was higher than that extracted from LSEM 550 (Fig 4).

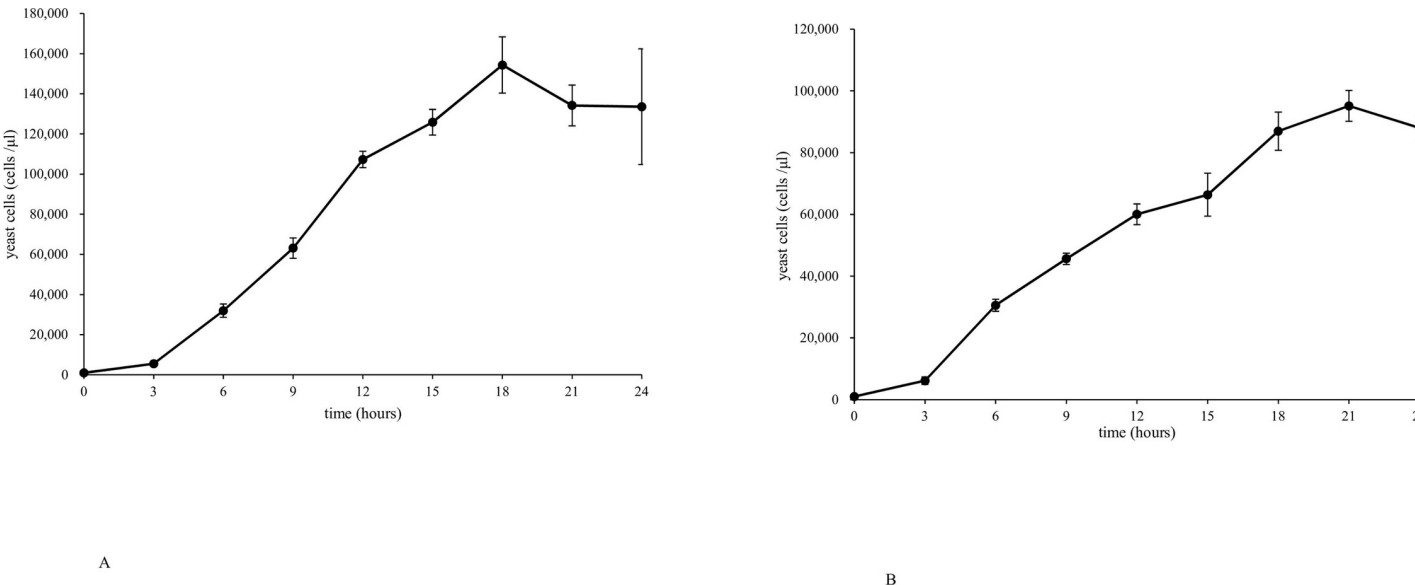

**Fig 1. Growth curves of *Candida albicans* NBRC1385 and LSEM 550 in Sabouraud's dextrose broth under aerobic conditions and shaking at 37˚C.** (A) The growth curve of NBRC 1385 was linear up to 18 h (approximately 170,000/μL) and then plateaued or slightly decreased. (B) The growth curve of LSEM 550 was roughly linear until 21 h (approximately 85,000 /μL) and then slightly decreased.

PFK 1 activity was determined over a pH range of 6.0–9.2 to investigate pH dependence and was the highest at pH 7.8–8.0 (Fig 4A). The regulation of PFK1 activity by F-6-P is shown in Fig 4B. When assayed with 0.5 mM ATP at pH 7.0, PFK1 showed a sigmoidal curve and higher activation with increasing concentrations of F-6-P. Additionally, the effects of ATP on the regulation of its activity were compared (Fig 4C and 4D). When assayed with 0.5 mM F-6-P at pH 7.0, PFK1 was activated by low concentrations (< 0.5 mM) of ATP (Fig 4C). Its activity was inhibited by millimolar concentrations of ATP, with nearly complete inhibition at 10 mM ATP (Fig 4D).

## Discussion

A prominent feature of some yeasts, such as *Saccharomyces cerevisiae*, is the rapid conversion of sugars to ethanol and carbon dioxide under both anaerobic and aerobic conditions. Under aerobic conditions, respiration, with oxygen as the final electron acceptor, is favored; nonetheless, some yeasts exhibit alcoholic fermentation until carbohydrate depletion. This phenomenon is known as the Crabtree effect [15,16]. *C. albicans* is considered a Crabtree-negative yeast; however, some studies have reported the Crabtree effect or similar effects in *C. albicans* [17–19]. Additionally, glucose metabolism has been suggested to be closely associated with morphological changes, such as yeast-mycelium transition [20] and germ tube production [21], which contribute to pathogenicity and infectivity.

In this study, we investigated the initial change in glucose catabolism in *C. albicans* under anaerobic or mitochondrial-impairment conditions by evaluating the changes in the expression of glycolysis-related enzymes in two strains of *C. albicans*: type strain NBRC 1385 and LSEM 550. Compared with NBRC 1385, LSEM 550 showed higher expression levels of many enzymes. Therefore, the high ability of LSEM 550 to rapidly respond to environmental changes could be one of pathogenicity.

Several studies [6,22,23] have reported that under short-term (3 h) anaerobic conditions, the mRNA expression of enzymes in the middle and last stages of glycolysis and alcoholic

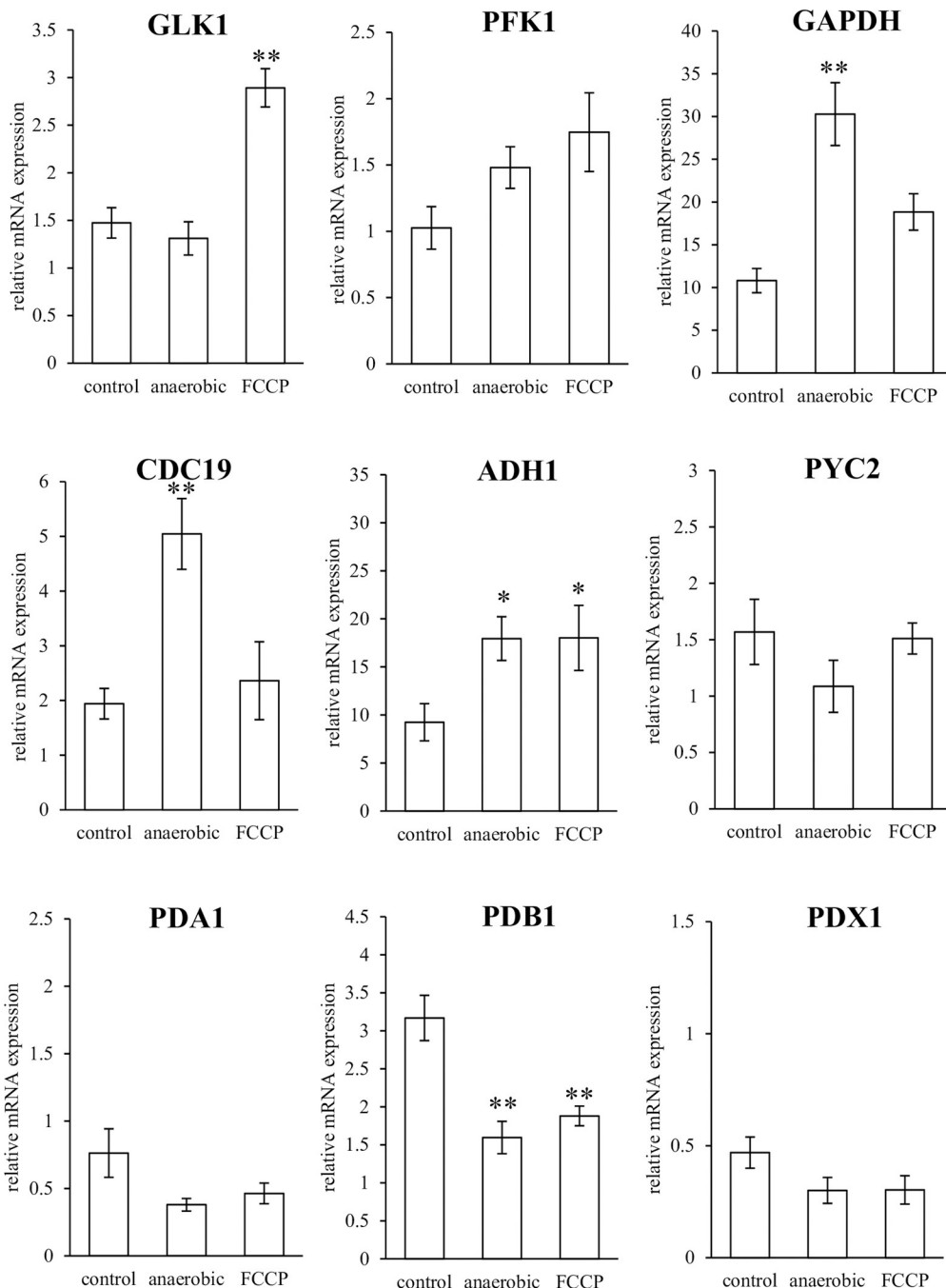

**Fig 2. mRNA expression of glycolysis-related enzymes in NBRC 1385.** The relative mRNA expression of glycolysis-related enzymes of NBRC 1385 in an anaerobic culture or after FCCP administration for 3 h was normalized to TBP1 mRNA expression. Data are presented as mean ± standard error (SE) and analyzed using Student's $t$-test. $*p < 0.05$, $**p < 0.01$. GLK1, glucokinase 1; PFK1, phosphofructokinase 1; GAPDH, glyceraldehyde 3-phosphate dehydrogenase; CDC19, pyruvate kinase; ADH1, alcohol dehydrogenase 1; PYC2, pyruvate carboxylase 2; PDA1, E1 α subunit; PDB1, E1 β subunit; PDX1, E3-binding protein; FCCP, carbonyl cyanide-p-trifluoromethoxyphenylhydrazone.

fermentation increases while that of enzymes involved in mitochondrial respiration decreases, which is consistent with our findings. Additionally, in our study, the uncouple of mitochondrial oxidative phosphorylation by FCCP administration showed similar results to those under

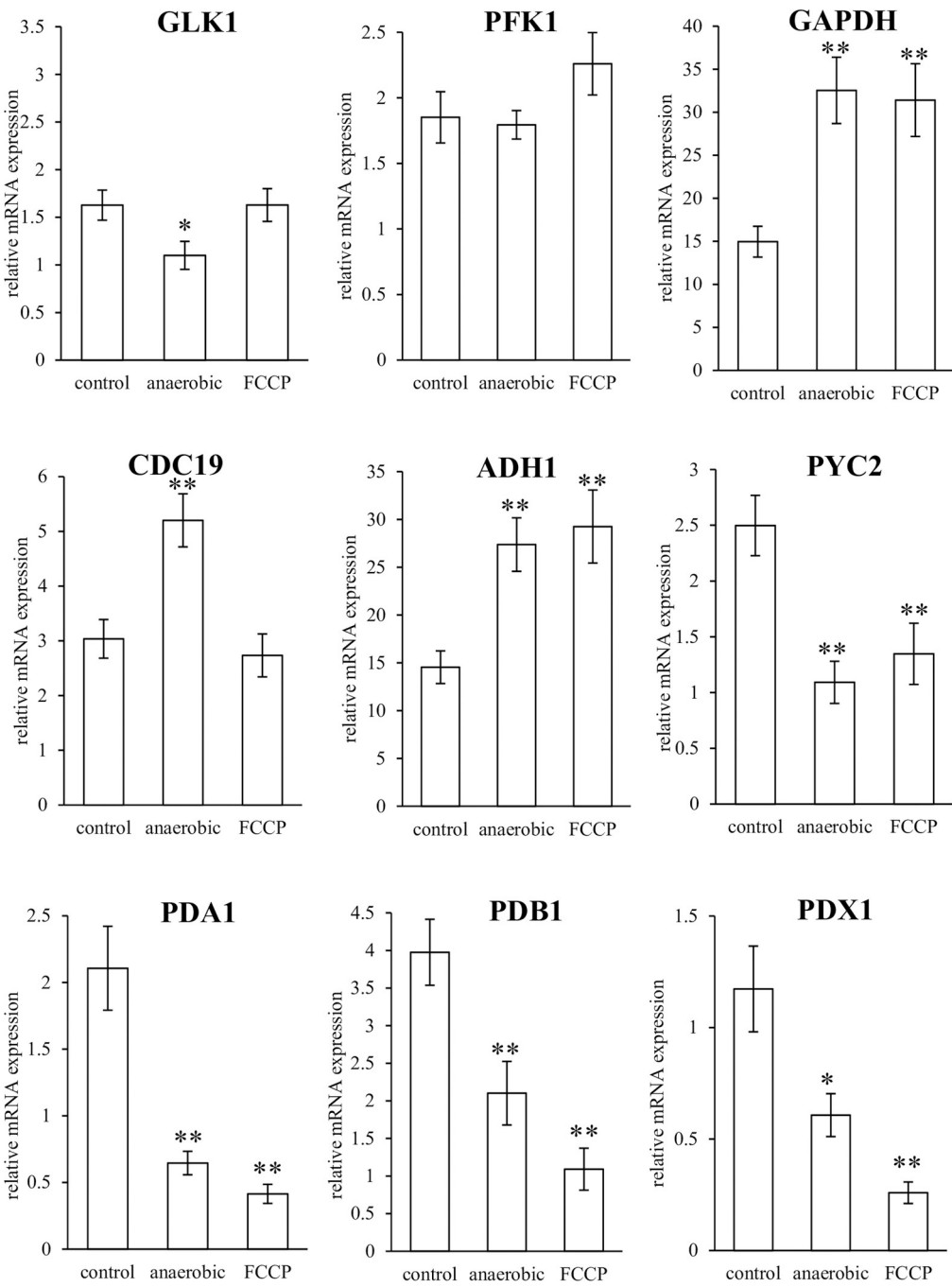

**Fig 3. mRNA expression of glycolysis-related enzymes in LSEM 550.** The relative mRNA expression of glycolysis-related enzyme LSEM 550 in an anaerobic culture or with FCCP administration for 3 h was normalized to TBP1 mRNA expression. Data are presented as mean ± standard error (SE) and analyzed using Student's t-test. $*p < 0.05$, $**$ $p < 0.01$. GLK1, glucokinase 1; PFK1, phosphofructokinase 1; GAPDH, glyceraldehyde 3-phosphate dehydrogenase; CDC19, pyruvate kinase; ADH1, alcohol dehydrogenase 1; PYC2, pyruvate carboxylase 2; PDA1, E1 α subunit; PDB1, E1 β subunit; PDX1, E3-binding protein; FCCP, carbonyl cyanide-p-trifluoromethoxy phenylhydrazone.

anaerobic conditions, except CDC19, the last enzyme of glycolysis; CDC19 mRNA expression was similar to that under aerobic conditions.

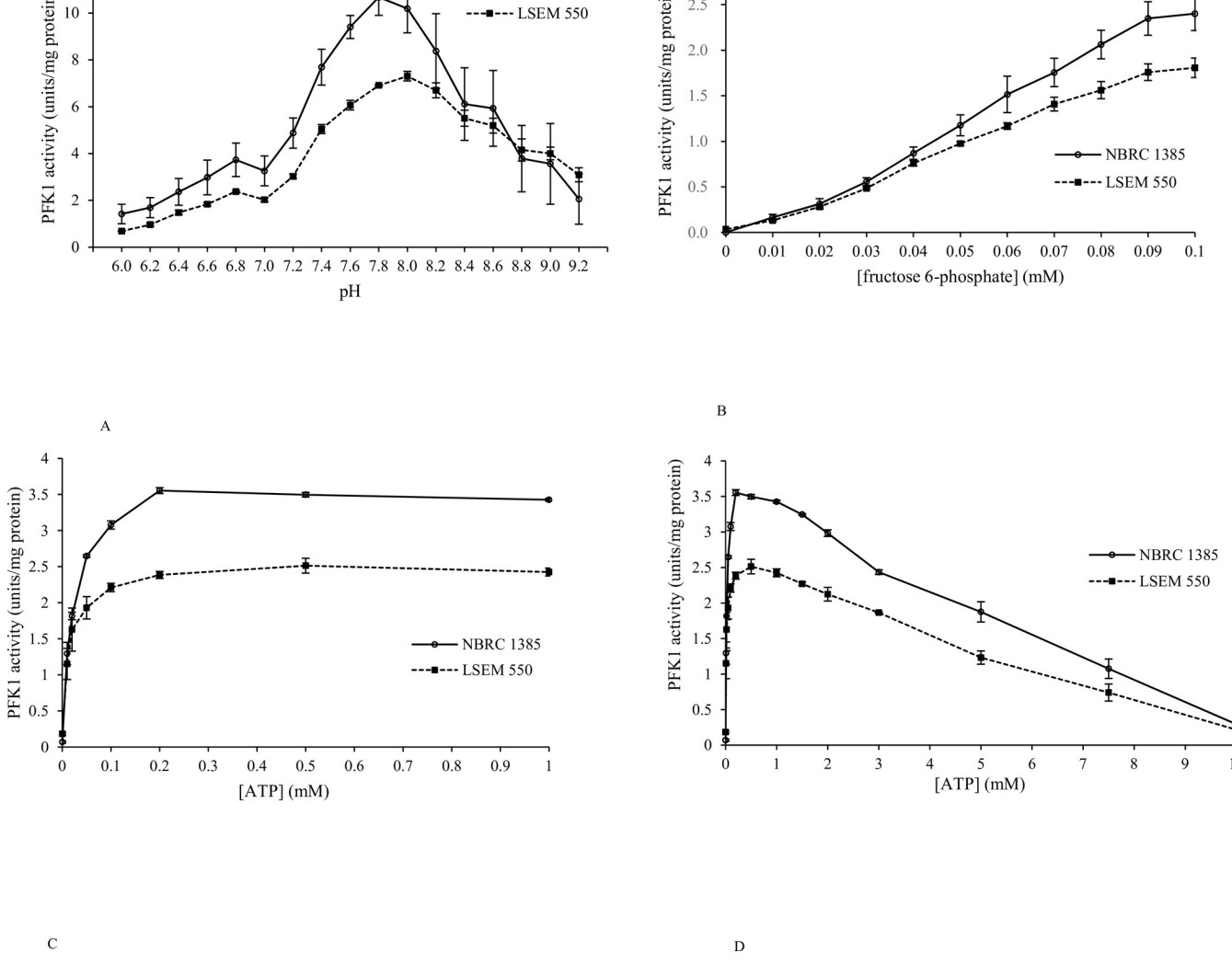

**Fig 4. Regulation of the activity of phosphofructokinase 1 (PFK1) under varying pH or substrate concentrations.** PFK1 activity was assayed in NBRC 1385 and LSEM 550 for pH dependency. (A) at various concentrations of fructose 6-phosphate (B) and ATP (C; 0–1 mM, D; 0–10 mM). Data are presented as mean ± standard error (SE).

PFK1, the most important regulatory enzyme in glycolysis, is regulated by allosteric mechanisms during morphological changes in *C. albicans* [24]. In this study, PFK1 showed a regulatory effect at various pH values and substrate concentrations; however, no significant change in its mRNA expression was observed. Additionally, PFK1 of *C. albicans* tolerates low pH and high ATP concentrations compared with that of mammals such as dogs [25]. Therefore, our results suggest that *C. albicans* obtains energy by carbohydrate catabolism even under harsh environments and survives in various parts of the host, including skin, oral cavity, gastrointestinal tract, and vagina.

The response of pathogenic fungi to various conditions is important for virulence [23,26]. A recent study reported that the fermentative enzyme ADH1 is required for the pathogenicity of *C. albicans* to mediate its effects on mitochondrial oxidative phosphorylation, but it does not affect glycolysis [27]. Additionally, the study indicated that mRNA expression of enzymes

associated with glucose catabolism changes rapidly in response to the inhibition of intramito-chondrial catabolism. In the present study, we elucidated a part of the initial regulation of gly-colysis and alcohol fermentation by *C. albicans* under different environmental conditions.

Alcohol produced by yeast exerts various toxic effects on the host. Our results suggested that the anaerobic environmental adaptation and intramitochondrial catabolic inhibition of *C. albicans* regulate glycolysis and increase alcohol production within the host body. Endogenous alcohol produced via anaerobic fermentation by *C. albicans* could cause auto-brewery syndrome.

The findings in this study suggest that the regulation of glycolysis could be the initial response to changes in yeast cell proliferation, morphology, infectivity, and pathogenicity under various environmental conditions. Further investigations are required to understand the environmental adaptations of *C. albicans* and elucidate the relationship between glycolysis and the pathogenicity of *C. albicans* in detail.

## Author Contributions

**Conceptualization:** Ken Okabayashi, Takanori Narita.

**Formal analysis:** Ken Okabayashi, Takanori Narita.

**Investigation:** Hitomi Ogawa, Yuto Hirai, Kureha Nagata, Yukiko Sato.

**Methodology:** Ken Okabayashi, Takanori Narita.

**Resources:** Kazuo Satoh, Koichi Makimura.

**Supervision:** Ken Okabayashi, Takanori Narita, Koichi Makimura.

**Visualization:** Ken Okabayashi, Takanori Narita.

**Writing – original draft:** Ken Okabayashi.

**Writing – review & editing:** Ken Okabayashi, Takanori Narita.

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
