## [Decision Letter · Decision Letter 0]

8 Feb 2023

PONE-D-22-29062Changes in the mRNAexpression of glycolysis-related enzymes of Candida albicans during inhibition of intramitochondrial catabolism under anaerobic conditionPLOS ONE

Dear Dr. Narita,

Thank you for submitting your manuscript to PLOS ONE. After careful consideration, we feel that it has merit but does not fully meet PLOS ONE’s publication criteria as it currently stands. Therefore, we invite you to submit a revised version of the manuscript that addresses the points raised during the review process. Please submit your revised manuscript by Mar 25 2023 11:59PM. If you will need more time than this to complete your revisions, please reply to this message or contact the journal office at plosone@plos.org. Please include the following items when submitting your revised manuscript:A rebuttal letter that responds to each point raised by the academic editor and reviewer(s). You should upload this letter as a separate file labeled 'Response to Reviewers'.A marked-up copy of your manuscript that highlights changes made to the original version. You should upload this as a separate file labeled 'Revised Manuscript with Track Changes'.An unmarked version of your revised paper without tracked changes. You should upload this as a separate file labeled 'Manuscript'.

We look forward to receiving your revised manuscript.

Kind regards,

Atul Vashist, PhD

Academic Editor

PLOS ONE

Journal Requirements:

Reviewers' comments:

Reviewer's Responses to Questions

**Comments to the Author**

1. Is the manuscript technically sound, and do the data support the conclusions?

Reviewer #1: Yes

Reviewer #2: Yes

Reviewer #3: No

2. Has the statistical analysis been performed appropriately and rigorously? 

Reviewer #1: Yes

Reviewer #2: No

Reviewer #3: Yes

3. Have the authors made all data underlying the findings in their manuscript fully available?

Reviewer #1: Yes

Reviewer #2: No

Reviewer #3: Yes

4. Is the manuscript presented in an intelligible fashion and written in standard English?

Reviewer #1: Yes

Reviewer #2: Yes

Reviewer #3: Yes

5. Review Comments to the Author

Reviewer #1: The manuscript is interesting, since it provides studies on the regulation of PFK1 as a regulator of glycolysis and possibly growth and physiology of C. albicans. As the authors propose, more studies are required, particularly the measurement of the actual activities of the different enzymes, as well as the levels of ATP under the different conditions used.

However, the experiments are a good start to go more deeply into the reagulation of metabolism, but more importanly, the infectivity of this yeast. Pehaps also, a comparative study with. S,. cerevisiae, as a model yeast. should be included in future experiments.

Reviewer #2: This study does not have any novelty, as we already knows about almost all the paradigm during anaerobic or hypoxia conditions in which cells produces frequent energy by glycolysis or HMP shunt and not through by OXPHOS mechanism in eukaryotic system. The other part of this study is to check only m-RNA expression are not only promising option to correct prediction of protein or enzyme expression, so for that authors should need to focus on proteomics and metabolomics domain for the better elucidation in anaerobic conditions in Candida Species.

Reviewer #3: Reviewer comments

Candida albicans is a major opportunistic pathogen of human capable of causing both superficial and systemic infections. Systemic C. albicans infection has emerged as a major cause of morbidity and mortality in immunocompromised patients. In the present article, the authors have studied the differential mRNA expression of enzymes in the glycolysis and alcohol fermentation to prove that C. albicans obtains energy by carbohydrate catabolism in the early phase of environmental change that helps them survive in various parts of the host. However, the claims made in the article about the pathogenesis potential of C. albicans and glycolysis require better validation.

The following are my comments

Major comments

1. The introduction and discussion should be revised and strengthened

2. Discussion: There are no experiments to prove that auto brewery syndrome is due to the ability of C. albicans to rapidly respond to environmental changes. Further it cannot be claimed by the results of comparing just two C. albicans.

3. The difference in gene expression among strains of candida in similar culture conditions needs to explained better with substantial experimental evidence.

Minor comments

1. Introduction: Line 45-46 should be rephrased

2. Introduction: Line 67-68 should be rechecked. Regulation of PFK1 was studied after partial purification of the enzyme.

3. Methods: Line 81-84. Mention the read out of the growth curve assay. Was the number of yeast cell counted using a hemocytometer or by determining the OD or CFU?

6. PLOS authors have the option to publish the peer review history of their article (what does this mean?). If published, this will include your full peer review and any attached files.

Reviewer #1: No

Reviewer #2: No

Reviewer #3: No

---

## [Author Response · Author response to Decision Letter 0]

19 Mar 2023

March 20, 2023

Dr. Atul Vashist  

Academic Editor

PLOS ONE

Dear Dr. Vashist,

Thank you for your comments and suggestions for the improvement of our manuscript (PONE-D-22-29062) titled “Changes in the mRNA expression of glycolysis-related enzymes of Candida albicans during inhibition of intramitochondrial catabolism under anaerobic condition”.

We have incorporated these suggestions in the revised manuscript (revised text is indicated in red for your convenience).

We have also provided point-by-point responses to all comments raised.

We hope that our manuscript is now suitable for publication in your esteemed journal.

Should you have any further comments or questions, please contact us.

Thank you for your consideration and we look forward to hearing from you.

Sincerely,

Takanori Narita

Department of Veterinary Medicine, College of Bioresource Sciences, Nihon University, 1866 Kameino, Fujisawa, Kanagawa, 252-0880, Japan

Tel: +81 466843848; Fax: +81 466843848

Email: narita.takanori@nihon-u.ac.jp

Reviewer #1: The manuscript is interesting, since it provides studies on the regulation of PFK1 as a regulator of glycolysis and possibly growth and physiology of C. albicans. As the authors propose, more studies are required, particularly the measurement of the actual activities of the different enzymes, as well as the levels of ATP under the different conditions used.

However, the experiments are a good start to go more deeply into the reagulation of metabolism, but more importanly, the infectivity of this yeast. Pehaps also, a comparative study with. S. cerevisiae, as a model yeast. should be included in future experiments.

Thank you for your comment and suggestion. In our upcoming study, we intend to analyze other enzyme activities related to glucose metabolism and elucidate the pathogenicity of C. albicans in comparison with S. cerevisiae as a non-pathogen.

Reviewer #2: This study does not have any novelty, as we already knows about almost all the paradigm during anaerobic or hypoxia conditions in which cells produces frequent energy by glycolysis or HMP shunt and not through by OXPHOS mechanism in eukaryotic system. The other part of this study is to check only m-RNA expression are not only promising option to correct prediction of protein or enzyme expression, so for that authors should need to focus on proteomics and metabolomics domain for the better elucidation in anaerobic conditions in Candida Species.

There are limited reports that have focused on the alcoholic fermentation of the pathogen C. albicans, though there are more reports on the important model organism S. cerevisiae. This study focused on the ability of alcoholic fermentation of C. albicans to induce auto-brewery syndrome. Our results indicate that the alcohol production of C. albicans is involved in pathogenicity. We intend to investigate alcohol production by C. albicans by incorporating proteomics and metabolomics. In the revised manuscript, we discussed the relationship between pathogenicity and alcohol production by C. albicans.

Reviewer #3: Candida albicans is a major opportunistic pathogen of human capable of causing both superficial and systemic infections. Systemic C. albicans infection has emerged as a major cause of morbidity and mortality in immunocompromised patients. In the present article, the authors have studied the differential mRNA expression of enzymes in the glycolysis and alcohol fermentation to prove that C. albicans obtains energy by carbohydrate catabolism in the early phase of environmental change that helps them survive in various parts of the host. However, the claims made in the article about the pathogenesis potential of C. albicans and glycolysis require better validation.

The following are my comments

Major comments

1. The introduction and discussion should be revised and strengthened

Thank you for your suggestion. The relationship between pathogenicity and alcohol fermentation by C. albicans was further discussed in the revised Introduction and Discussion

2. Discussion: There are no experiments to prove that auto brewery syndrome is due to the ability of C. albicans to rapidly respond to environmental changes. Further it cannot be claimed by the results of comparing just two C. albicans.

We have revised the Discussion accordingly.

3. The difference in gene expression among strains of candida in similar culture conditions needs to explained better with substantial experimental evidence.

We have added text on the comparison of gene expression between the two strains in the revised Results “Quantitative reverse transcriptase PCR”.

Minor comments

1. Introduction: Line 45-46 should be rephrased

We have revised the text accordingly.

2. Introduction: Line 67-68 should be rechecked. Regulation of PFK1 was studied after partial purification of the enzyme.

We have revised the text accordingly.

3. Methods: Line 81-84. Mention the read out of the growth curve assay. Was the number of yeast cell counted using a hemocytometer or by determining the OD or CFU?

We used a hemocytometer to count the number of yeast cells. We have indicated that in the revised manuscript.

---

## [Decision Letter · Decision Letter 1]

29 Mar 2023

Changes in the mRNAexpression of glycolysis-related enzymes of Candida albicans during inhibition of intramitochondrial catabolism under anaerobic condition

PONE-D-22-29062R1

Dear Dr. Takanori Narita,

We’re pleased to inform you that your manuscript has been judged scientifically suitable for publication and will be formally accepted for publication once it meets all outstanding technical requirements.

Kind regards,

Atul Vashist, PhD

Academic Editor

PLOS ONE

Additional Editor Comments (optional):

Authors are requested to please address the one minor comment made by Reviewer 1 and accordingly incorporate this in the manuscript. 

Reviewers' comments:

Reviewer's Responses to Questions

**Comments to the Author**

1. If the authors have adequately addressed your comments raised in a previous round of review and you feel that this manuscript is now acceptable for publication, you may indicate that here to bypass the “Comments to the Author” section, enter your conflict of interest statement in the “Confidential to Editor” section, and submit your "Accept" recommendation.

Reviewer #1: All comments have been addressed

Reviewer #2: All comments have been addressed

Reviewer #3: All comments have been addressed

2. Is the manuscript technically sound, and do the data support the conclusions?

Reviewer #1: Yes

Reviewer #2: Yes

Reviewer #3: Partly

3. Has the statistical analysis been performed appropriately and rigorously? 

Reviewer #1: Yes

Reviewer #2: Yes

Reviewer #3: Yes

4. Have the authors made all data underlying the findings in their manuscript fully available?

Reviewer #1: Yes

Reviewer #2: Yes

Reviewer #3: Yes

5. Is the manuscript presented in an intelligible fashion and written in standard English?

Reviewer #1: Yes

Reviewer #2: Yes

Reviewer #3: Yes

6. Review Comments to the Author

Reviewer #1: There is only one minor comment: FCCP does not exactly inhibit oxidative phosphorylation, but uncouples it

Reviewer #2: (No Response)

Reviewer #3: The revised manuscript can be accepted.

Candida albicans is a major opportunistic pathogen of human capable of causing both superficial and systemic infections. Systemic C. albicans infection has emerged as a major cause of morbidity and mortality in immunocompromised patients. In the present article, the authors have studied the differential mRNA expression of enzymes in the glycolysis and alcohol fermentation to prove that C. albicans obtains energy by carbohydrate catabolism in the early phase of environmental change that helps them survive in various parts of the host.

7. PLOS authors have the option to publish the peer review history of their article (what does this mean?). If published, this will include your full peer review and any attached files.

Reviewer #1: **Yes: **Antonio Peña

Reviewer #2: **Yes: **Vivek Singh

Reviewer #3: No

---

## [Editor Report · Acceptance letter]

10 Apr 2023

PONE-D-22-29062R1 

Changes in the mRNA expression of glycolysis-related enzymes of *Candida albicans* during inhibition of intramitochondrial catabolism under anaerobic condition 

Dear Dr. Narita:

I'm pleased to inform you that your manuscript has been deemed suitable for publication in PLOS ONE. Congratulations! Your manuscript is now with our production department. 

Kind regards, 

on behalf of

Dr. Atul Vashist 

Academic Editor

PLOS ONE